# An Experimental Detection of Distributed Denial of Service Attack in CDX 3 Platform Based on Snort

**DOI:** 10.3390/s23136139

**Published:** 2023-07-04

**Authors:** Chin-Ling Chen, Jian Lin Lai

**Affiliations:** Department of Information Management, National Pingtung University, Pingtung 900, Taiwan; jason971128@gmail.com

**Keywords:** DDoS, intrusion detection system, internet security, cloud computing

## Abstract

Distributed Denial of Service (DDoS) attacks pose a significant threat to internet and cloud security. Our study utilizes a Poisson distribution model to efficiently detect DDoS attacks with a computational complexity of O(*n*). Unlike Machine Learning (ML)-based algorithms, our method only needs to set up one or more Poisson models for legitimate traffic based on the granularity of the time periods during preprocessing, thus eliminating the need for training time. We validate this approach with four virtual machines on the CDX 3.0 platform, each simulating different aspects of DDoS attacks for offensive, monitoring, and defense evaluation purposes. The study further analyzes seven diverse DDoS attack methods. When compared with existing methods, our approach demonstrates superior performance, highlighting its potential effectiveness in real-world DDoS attack detection.

## 1. Introduction

Network attacks, especially Distributed Denial of Service (DDoS) attacks, pose significant concerns due to their severity. Over the past two decades, DDoS attacks have caused considerable disruptions across various industries, leading to major catastrophes and service interruptions. In September 2017, Google Cloud withstood the largest recorded DDoS attack, peaking at a staggering 2.5 Tbps [1]. In February 2018, GitHub experienced a 1.3 Tbps attack [2], while AWS successfully defended against a 2.3 Tbps DDoS assault in February 2020 [3]. These incidents underscore the crucial importance of proactive defensive measures against network attacks to effectively neutralize threats to information security. Without suitable countermeasures, the consequences of such attacks could be catastrophic.

A host-based intrusion detection system (IDS), Snort, is capable of identifying unusual network traffic. The study [4] explores Snort’s application in analyzing bandwidth traffic for intrusion detection and prevention systems, and it presents defensive strategies for two common types of attacks. This work thoroughly discusses the IDS’s implementation, structure, and intrusion detection method. In [5], a DDoS mitigation approach leveraging Snort for DDoS detection within a Eucalyptus private cloud setup is proposed. While corporate networks often implement multiple protection measures, protection in home environments remains relatively weak, frequently due to home users’ lack of security-risk awareness. A cost-effective intrusion protection system, based on Snort and using the Raspberry Pi 3 B+ model, is proposed in [6]. This system uses the TaZmen Sniffer Protocol (TZSP) to analyze network traffic and calculate periodic hash values with the SHA3 algorithm, offering an affordable solution to enhance home network security.

Slow HTTP DoS attacks pose a significant threat to HTTP servers. In response to the challenge of distributed slow HTTP DoS attacks, a defense mechanism is proposed in [7]. This mechanism thwarts potential attack connections by monitoring the quantity and duration of connections per IP address. The advent of Information-Centric Networks (ICNs) has ushered in a new paradigm for content distribution, access, and retrieval. However, these networks remain susceptible to DDoS attacks. To address this issue, a robust mechanism, GET Message Logging-based Filtering (GMLF), designed to combat path identifier-based attacks targeting ICNs, is introduced in [8]. The mechanism employs Bloom filter logging to store incoming GET messages, validates related content messages, and filters packets from malicious hosts.

DDoS attacks pose a significant risk to cloud-based systems, having the potential to cause substantial financial disruption. Current defense strategies often overlook the sophisticated tactics employed by attackers, particularly those exploiting the elasticity and multi-tenant features of the cloud, and fail to account for the constraints of the cloud system’s finite resources. A real-time detection mechanism for TCP-based DDoS attacks is introduced in [9]. This mechanism utilizes two decision tree classifiers to select effective features from TCP traffic and distinguish between malicious and normal traffic. A cooperative fair rate adjustment mechanism, treating the attacks as rate management and congestion control issues, is proposed in [10] to counter DDoS attacks. This solution presents a decentralized defense architecture, featuring an anomaly detection mechanism for identifying attacks, an early detection mechanism, and a feedback system between autonomous systems (ASes). The study further introduces secure, private, authenticated channels to manage the feedback process and an active resource management mathematical model. A dual-pronged solution is proposed in [11]. The first component enhances the hypervisor’s ability to form robust trust relationships with guest Virtual Machines (VMs). The second component involves designing a trust-based maximin game. The solution to this game offers strategic advice to the hypervisor, enabling it to dynamically determine the most beneficial detection load distribution among VMs. DDoS attack detection in cloud security is essential for ensuring uninterrupted access to cloud resources. Machine Learning (ML)-based IDSs have shown promise in handling network incidents, including DDoS attacks. In this context, feature selection in ML classification plays a critical role. An ensemble framework for feature selection methods (EnFS) is presented in [12]. This framework combines seven well-known feature selection methods using a majority voting (MV) technique. This approach demonstrates higher accuracy and fewer false alarms compared to existing methods, indicating the effectiveness of EnFS in enhancing IDS performance. The study [13] utilizes various ML classifiers to detect and classify attack traffic and normal traffic. They employ five common feature-selection methods on the NSL KDD dataset. Their proposed hybrid method demonstrated the highest detection rate compared to existing approaches. The study [14] outlines various cloud defense mechanisms, including prevention, detection, and mitigation techniques, and emphasizes the challenges in distinguishing between high traffic due to a DDoS attack and legitimate high traffic.

The use of blockchain technology to enhance the mitigation of DDoS attacks is proposed in [15]. By leveraging the smart contracts of the Ethereum blockchain, an Intrusion Prevention System (IPS) can share information about attack origins or blacklisted IPs with other IPSs, thereby streamlining the mitigation process. Software Defined Networking (SDN), a pivotal enabling technology in the current landscape, offers a novel and robust network architecture, which allows dynamic operation of different services on a common network infrastructure. The study discusses the vulnerabilities of SDN and proposes the selection of specific attack attributes to identify those most influential to anomaly detection [16]. These selected attributes are used to train the model, enhancing its performance while reducing computational costs. The research concludes with detailed analyses and simulation results, revealing the primary attributes and their levels of impact on different attacks. In [17], an analysis of DDoS threats and a review of innovative defense mechanisms are presented. The study extensively discusses performance metrics commonly used for evaluating these defense strategies. The paper concludes with a list of common DDoS attack tools and open challenges.

The Cyber Defense eXercise (CDX) platform [18], under the guidance of the National Science Council and implemented by the National Center for High-Performance Computing (NCHC), Taiwan, is part of the “Information Security Open Data Platform Development and Malware Knowledge Base Maintenance (II)” project. This platform adopts a cloud service architecture for its planning and design, primarily to overcome the limitations of traditional cyber defense platforms due to hardware and software constraints and issues related to ease of management and use. Leveraging a virtualized architecture, the CDX platform demonstrates the feasibility of swiftly deploying cyberattack and defense exercise scenarios. It provides an environment conducive to multi-player and multi-scenario exercises simultaneously, and can also simulate real network environments for related research in cyberattack and defense techniques. It enables participants to familiarize themselves with and master past information security incidents, learning from these to hone their detection and analysis skills in the realm of information security.

In this paper, we leverage the Poisson distribution model to detect potential DDoS attacks. Unlike Machine Learning (ML) detection algorithms, our research does not require training time. Instead, it only needs the creation of several Poisson distribution models for legitimate traffic during preprocessing. Then, by comparing the model with the data collected from the network, the presence of attack traffic can be determined. We conduct experiments simulating seven different DDoS attack methods and defense techniques on virtual machines hosted on a cloud platform. Packet analysis tools and performance monitoring utilities are employed to detect whether the system is under attack. These performance monitoring tools continuously collect data, enabling a comparison of variations in attack methods and their respective impacts on the system. We utilize Snort as an intrusion detection system (IDS) in this study to identify malicious attack packets and record alarm logs. Timely detection during an attack allows for the implementation of strategies to mitigate or defend against the assault. Finally, we compare our research with existing methods. The results reveal that our study achieves superior performance.

The remainder of this paper is organized as follows: Section 2 presents the methodology of the proposed scheme. Section 3 illustrates the system architecture and corresponding experimental results. Finally, we conclude this paper and outline potential future work.

## 2. Research Methodology

We assume that the arrivals from legal traffic sources follow a Poisson process. Let us consider *K* independent sources, where each source, *k*, is a Poisson process with a rate of *λ_k_*, generating normal packets per second. A merged arrival stream is formed by amalgamating the inputs from all *K* sources. Let us consider the following argument as true [19].(1)The merged stream maintains the Poisson property, characterized by a parameter *λ*, which is the sum of rates from all individual sources, i.e., *λ* = *λ*_1_ +*λ*_2_ + … + *λ_k_*.(2)Suppose *p_i_* is the probability that a packet from the merged stream is assigned to the *i*th sub-stream. Given an overall arrival rate of *λ* packets per second, the *i*th sub-stream also follows a Poisson process with a rate of *λp_i_*.(3)Define *X_j_* as a sequence of identically distributed, mutually independent Bernoulli random variables, such that *P* [*X_j_* = 1] = *p* and *P* [*X_j_* = 0] = 1 − *p*. The sum is *S_N_* = *X*_1_ + … + *X_N_*, for a random number *N* of these variables *X_j_*, where *N* follows a Poisson distribution with a mean *λ. S_N_* also adheres to a Poisson distribution, but with a mean of *λp*.(4)In the *M*^[*x*]^/*M*/1 model, the actual number of arrival packets is a random variable, *X*, where *x* ∈ *X*, with a corresponding probability of *c_x_*. *λ_x_* is the arrival rate of the Poisson process for batches of size *X*. Therefore, we can express *c_x_* = *λ_x_*/*λ*, where *λ* represents the compound arrival rate of all batches. Clearly, *λ* equals the summation of all individual rates, or *λ* = ∑*_i_*_=1_^∞^*λ_i_*.

In a DDoS attack, the packet arrival rate might suddenly increase significantly. We can compare this behavior with the Poisson distribution model to detect possible DDoS attacks. From the arguments mentioned above, we may induce that the burst arrivals from DDoS attacker sources do not follow the Poisson process. Figure 1 depicts a flowchart of using Poisson distribution to detect and prevent DDoS attacks, which can be described as follows.
Step 1: Collect Baseline Data. Collect network traffic data in non-attack scenarios to obtain the packet arrival rate under normal conditions.Step 2: Establish Poisson Model. Based on the arrival rate under normal conditions, establish a Poisson distribution model.Step 3: Real-time Monitoring. Monitor network traffic in real-time and calculate the arrival rate in a specific time range.Step 4: Anomaly Detection. If the arrival rate in a specific time period is significantly higher than the arrival rate expected by the Poisson model, a DDoS attack may exist. Specifically, if the actual arrival rate exceeds the predicted 95% confidence interval, an alert will be generated within the system, prompting the subsequent initiation of procedures to block the suspicious IP address.

The computational complexity of our algorithm using Poisson distribution for anomaly detection in DDoS attack scenarios primarily depends on two aspects:(1)Data Collection and Preprocessing: This involves collecting and processing network traffic data, which typically is a constant time operation, O(1), for each packet, but in total it is O(*n*), where *n* is the number of packets.(2)Anomaly Detection: This part involves calculating the Poisson distribution parameters, calculating the expected arrival rate for each time period, and comparing it with the observed rate. These are generally arithmetic operations, and if performed for each packet, the complexity would be O(*n*).

So, the overall time complexity of the algorithm is linear, i.e., O(*n*), assuming that we process each packet once.

Poisson distribution is a commonly used statistical model for representing the occurrence of events over a specified interval of time. It can be especially useful in detecting DDoS attacks due to the following reasons:(1)Modeling Event Frequency: DDoS attacks usually involve a sudden increase in network traffic over a short period of time, which deviates significantly from normal behavior. The Poisson distribution can effectively model this because it describes the probability of a given number of packet arrivals occurring in a fixed interval of time.(2)Simplicity and Efficiency: The Poisson model is relatively simple to understand and implement, and it requires fewer computational resources compared to more complex models. This allows for efficient real-time analysis, which is critical in DDoS detection where timely response is of the essence.(3)Independence of Events: The Poisson distribution assumes that each event is independent of the others. This assumption aligns well with certain types of DDoS attacks, where each request (or packet) can be considered independent.(4)Usefulness in Anomaly Detection: A significant deviation from the expected number of packet arrivals, based on the Poisson distribution, can be considered an anomaly. Therefore, using the Poisson distribution, we can develop a threshold for what constitutes “normal” behavior, and anything exceeding that could be flagged as a potential DDoS attack.

## 3. System Architecture and Experiment Results

In the proposed system architecture, we deploy four virtual machines (VMs) on the CDX 3.0 cloud platform [18]. Figure 2 illustrates the configuration and setup of these VMs on the CDX platform. The setup involves initializing various types of operating systems (such as Windows, Linux, etc.) with distinct roles (like attackers, victims, etc.). It also encompasses the configuration of necessary resources, including the number of CPU cores, memory size, and network interface setup. The VMs are typically configured to closely mirror real-world systems. We can model the precise nature of background traffic and attack patterns on this platform. The first VM, playing the role of the attacker, is built on Ubuntu with KALI installed. KALI generates multiple private IP addresses, simulating the behavior of cloned botnets. To mimic DDoS attacks, we employ Hping3 within the KALI VM. As the number of bots increases, the flow of traffic correspondingly intensifies. The second key component of our system is the intrusion detection system (IDS), which is based on Ubuntu and installed with Snort. The KALI VM generates a flood of network traffic directed towards a specific target, in this case, Snort, within the platform. The third component of the system utilizes Wireshark for the analysis of incoming packets. Wireshark offers an in-depth understanding of the network’s activities at the protocol level and is capable of decoding and analyzing protocols from the network layer up to the application layer. In the case of a security incident, Wireshark can be utilized to inspect packet payloads for potential malicious content or patterns that align with known attack signatures. The final component is tasked with monitoring the attacker’s traffic flow. It employs PRTG, installed on an Ubuntu-based system. PRTG is used to monitor crucial network performance indicators such as bandwidth usage, latency, and packet loss. During a DDoS attack scenario, PRTG is capable of sending an alert in response to an unusual spike in incoming traffic, enabling a prompt reaction. Figure 3 depicts the experimental architecture, and Table 1 provides a related description of the experiment.

Here are the details on how to implement the Poisson distribution model in Snort:(1)Data Collection: Regular, non-attack network traffic data should be collected over a specified period of time. The data should be comprehensive enough to cover different times of the day and different days of the week. We focus on the number of arriving packets per minute.(2)Model Setup: Using the collected data, calculate the average rate (*λ*) of packet arrivals per minute. This will serve as a parameter for our Poisson distribution.(3)Threshold Establishment: Define a threshold at which the rate of packet arrivals becomes anomalous. This threshold should be set high enough to minimize false positives, but low enough to catch actual attacks. It could be determined based on the statistical properties of the Poisson distribution, such as a number of standard deviations above the mean. In our study, we set it at two standard deviations above the mean.(4)Real-Time Monitoring and Anomaly Detection: During real-time operation, incoming traffic should be divided into the same time units (minutes) used during model training. For each time unit, calculate the number of arriving packets. Use the Poisson probability mass function to determine the likelihood of observing that number of packet arrivals, given the average rate *λ*. If the calculated probability falls below a certain threshold, or equivalently, if the number of arriving packets exceeds the established threshold, the traffic should be flagged as a potential DDoS attack.(5)Prevention: Once an anomalous traffic pattern has been detected, the system can trigger an alarm for predefined prevention strategies, blocking the suspicious IP address.

In the experiment, we generated seven types of DDoS attacks using Hping3 to test Snort’s rules and monitor attack traffic. The seven types of DDoS attacks are: TCP SYN Flood, UDP Flood, TCP FIN Flood, TCP RST Flood, PUSH and ACK Flood, ICMP Flood, and Smurf attack. When Snort detects attack traffic, it stores the relevant information in an alert file. The experimental procedures can be described as follows.
TCP SYN FloodAttack command:
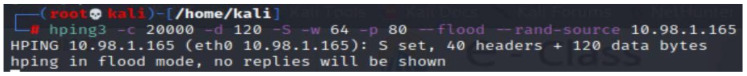
Snort rule:
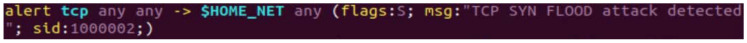
Part of alert file:
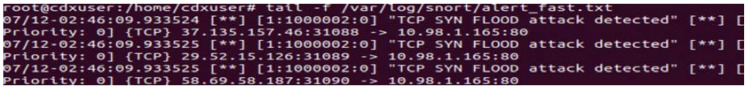
UDP FloodAttack command:
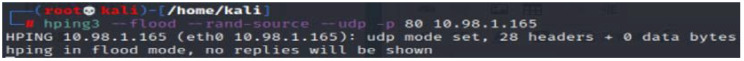
Snort rule:
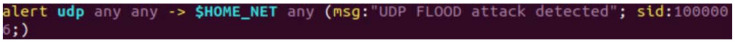
Part of alert file:
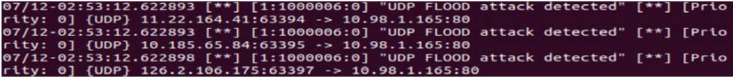
TCP FIN FloodAttack command:
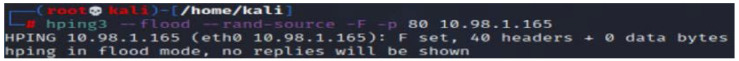
Snort rule:
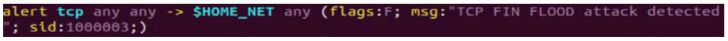
Part of alert file:
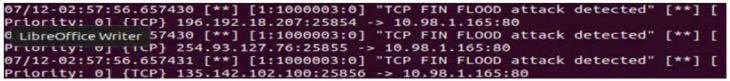
TCP RST FloodAttack command:
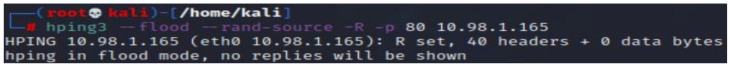
Snort rule:
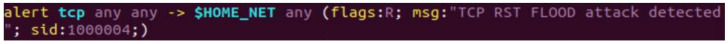
Part of alert file:
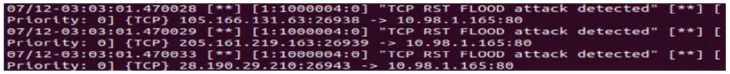
PUSH and ACK FloodAttack command:
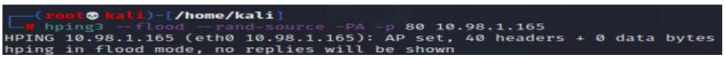
Snort rule:
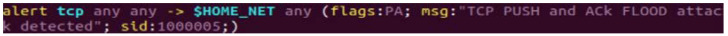
Part of alert file:
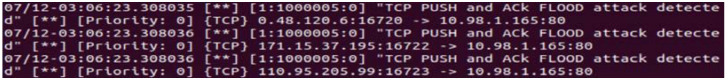
ICMP FloodAttack command:
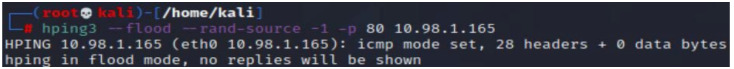
Snort rule:
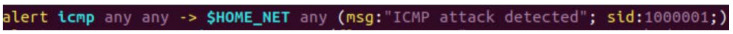
Part of alert file:
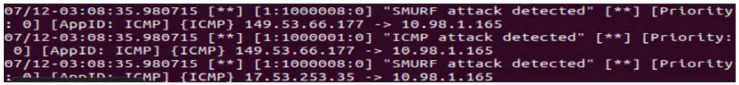
Smurf attackAttack command:
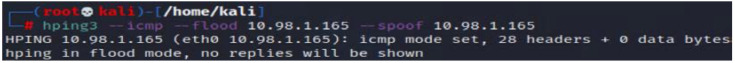
Snort rule:
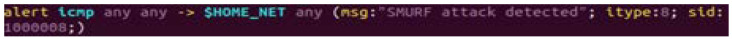
Part of alert file:
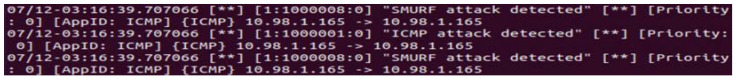


### 3.1. TCP SYN Flood

Figure 4, Figure 5 and Figure 6 represent CPU loading, available memory, and incoming flow under a TCP SYN flood, respectively. We observed all changes occurring at 09:24, the time the system was subjected to the TCP SYN flood attack. The CPU loading rises sharply to a maximum of 52%. Available memory, due to the impact of the attack, gradually drops to a minimum of 8.6%, and the network traffic increased significantly to a maximum of 529 Mbit/s.

### 3.2. UDP Flood

Figure 7, Figure 8 and Figure 9 reveal performance changes that occurred at 01:25, when the system was subjected to a UDP flood. CPU loading grew sharply to a maximum of 27%, and available memory gradually decreased to the minimum of 20.3%. Incoming attack flow increased dramatically to the highest rate of 211 Mbit/s.

### 3.3. TCP FIN Flood

Figure 10, Figure 11 and Figure 12 showcase the system’s performance under a TCP FIN flood, with all changes observed at 03:15. The CPU load rose sharply to a maximum of 30% due to the impact of the attack. The available memory gradually dropped to a minimum of 30.8%, and the network traffic surged a maximum of 213 Mbit/s.

### 3.4. TCP RST Flood

Figure 13, Figure 14 and Figure 15 respectively, indicate performance changes under a TCP RST flood, with changes noticed at 03:44. The CPU load advanced sharply to a maximum of 28%, while due to the impact of the attack, the available memory gradually decreased to a minimum of 28.7%. At the same time, incoming attack traffic grew dramatically to a maximum of 211 Mbit/s.

### 3.5. PUSH/ACK Flood

Figure 16, Figure 17 and Figure 18 depict performance changes under a PUSH/ACK flood attack that started at 10:05. The CPU load ascended quickly to a maximum of 42%, while the available memory descended slowly to a minimum of 25.4%. The attack traffic climbed greatly to a maximum of 206 Mbit/s.

### 3.6. ICMP Flood

Figure 19, Figure 20, Figure 21 and Figure 22 depict system alterations at 10:12 during an ICMP flood. Figure 21 illustrates downtime, representing the period when the system was non-operational. A downtime of 100% signifies a network overwhelmed by PING packets, preventing any further operation.

### 3.7. Smurf Attack

Figure 23, Figure 24, Figure 25 and Figure 26 illustrate the system’s performance shifts during a Smurf attack initiated at 03:37. The CPU load spiked to 28%, available memory decreased to a minimum of 22.6%, and network traffic surged to a peak of 207 Mbit/s. Figure 26 reveals that the packet loss rate, while remaining at 0% under normal traffic, fluctuated to signal an impending attack.

### 3.8. PING Test

Using the PING command, we tested response time and packet loss rate when the victim was under different types of attacks. The monitoring end (10.99.192.3) uses the PING command to send 100 ECHO_REQUEST packets to the victim end (10.99.192.4), repeating 10 times. According to the experimental results, it can be found that both ICMP flood and Smurf attack have the highest packet loss rates (100%). The reason is that the attack characteristics of both misuse PING packets, making the victim (10.99.192.4) full of EHCO_REQUEST. These ECHO_REPLY cannot be returned to 10.99.192.3 normally, resulting in 100% packet loss rate. Table 2 displays the comparison of packet loss rates under different attack scenarios.

In Table 3, we used PING packets to measure response time. Both ICMP flood and Smurf attacks resulted in 100% packet loss rates, which meant response time could not be measured. Both types of attack are not applicable to statistics of response time. Compared to the other five types of attack, the TCP SYN flood attack has the longest network response time, which is 0.39 ms longer than normal traffic.

We summarized the aforementioned experimental results in Table 4. Table 5 unveils the performance disparities before and after the attacks. As evidenced by Table 5, the TCP SYN flood attack influences CPU loading the most, marking a 34.68% increase post-attack. The ICMP flood attack contributes to a 15.48% rise in memory utilization, while the other kinds of attacks do not exhibit any considerable impact. The TCP FIN flood attack notably affects traffic flow, generating a surge of 69.52 Mbit/s. The packet loss rate triggered by the ICMP flood and Smurf attacks is substantial, reaching 100%. As every response packet is lost, it is impossible to measure any response time. The TCP SYN flood results in the longest response time, clocking in at 0.39 ms longer than standard traffic. It is important to pay particular attention to the memory utilization in Table 5. The memory utilization actually declines following a TCP-related attack, owing to Windows 10’s protective mechanism that filters out half-open or abnormal TCP connections.

Next, we benchmark our method against existing DDoS detection approaches, namely, the Radial Basis Function (RBF) Network, Support Vector Machine (SVM), Bagging, and J48 Decision Tree [20]. We evaluate their performance in terms of accuracy, false positives, and training time. For the experiment, we selected the LIBSVM package [21] with an RBF kernel. Table 6 shows that our method, RBF, and SVM outperform others in accuracy and false-positive rate. While RBF matches SVM in results, it demands a much longer training time. Our method, however, negates training time by establishing one or more Poisson models for legally arriving packets during preprocessing.

## 4. Conclusions and Future Works

In this study, we propose an innovative detection mechanism that utilizes the Poisson distribution model to identify potential DDoS attacks. This model has a computational complexity of O(*n*), offering an efficient alternative to more resource-intensive methods. We utilize Snort as an intrusion detection system and Wireshark and PRTG for analyzing and monitoring the impact of attacks. Our findings reveal that different types of attacks have varied impacts on system performance. A TCP SYN flood attack has a more significant effect on CPU utilization and response time than other types of attacks. Conversely, memory utilization is most impacted by ICMP flood attacks. Both ICMP flood and Smurf attacks have the greatest impact in terms of packet loss rate among all the attack types studied. Experimental results confirm the effectiveness of this approach. When compared with existing detection mechanisms, our model demonstrated superior performance metrics, indicating its potential usefulness in real-world applications. Thus, this research offers a novel and effective approach to the ongoing challenge of DDoS attack detection.

The experiments were conducted on the virtual machines of the CDX 3.0 cloud platform, which may not perfectly replicate a real network environment. It is important to note that the proposed method might overreact to sudden normal traffic changes, leading to false positives. Therefore, it may be necessary to use it in combination with other detection methods to ensure accuracy and completeness. Moreover, the actual computational requirements can be significantly influenced by factors such as the volume of network traffic and the granularity of the time periods. Specifically, the smaller the time periods, the more computations required. Additionally, the space complexity will be affected by the volume of historical data required to establish the baseline model.

## Figures and Tables

**Figure 1 sensors-23-06139-f001:**
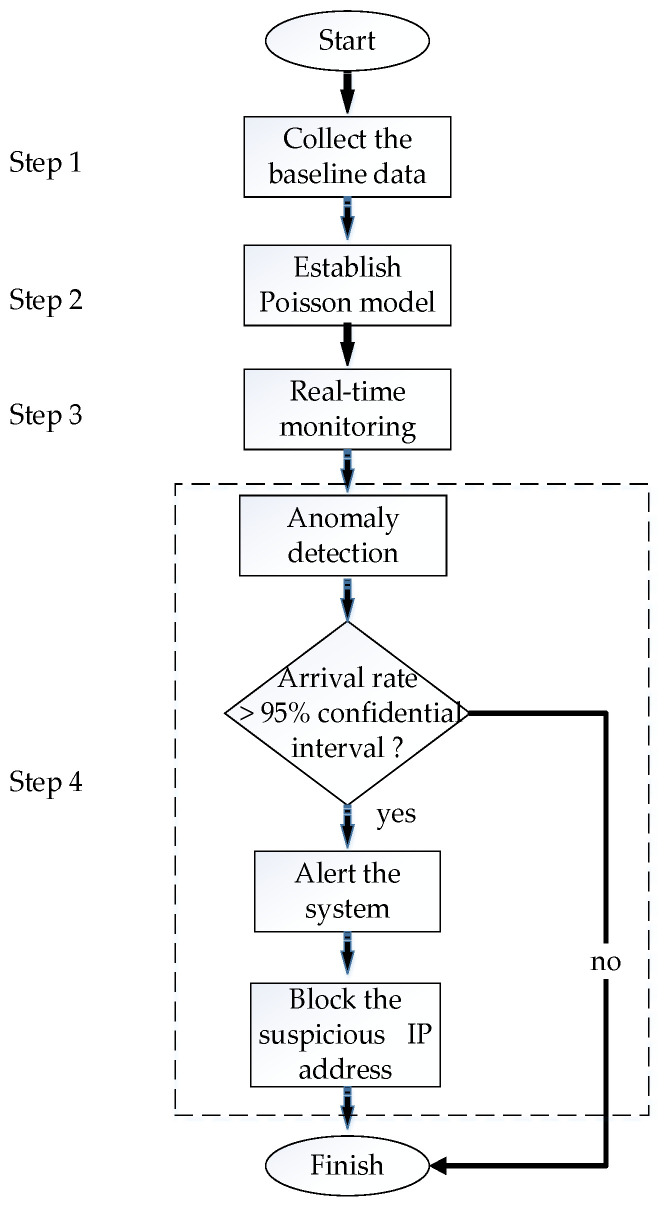
Flowchart of using Poisson distribution to detect and prevent DDoS attacks.

**Figure 2 sensors-23-06139-f002:**
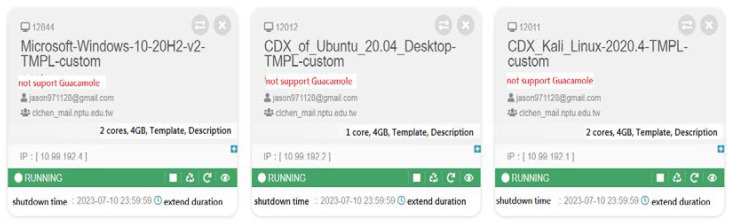
Configuration and setup of the virtual machines on the CDX platform.

**Figure 3 sensors-23-06139-f003:**
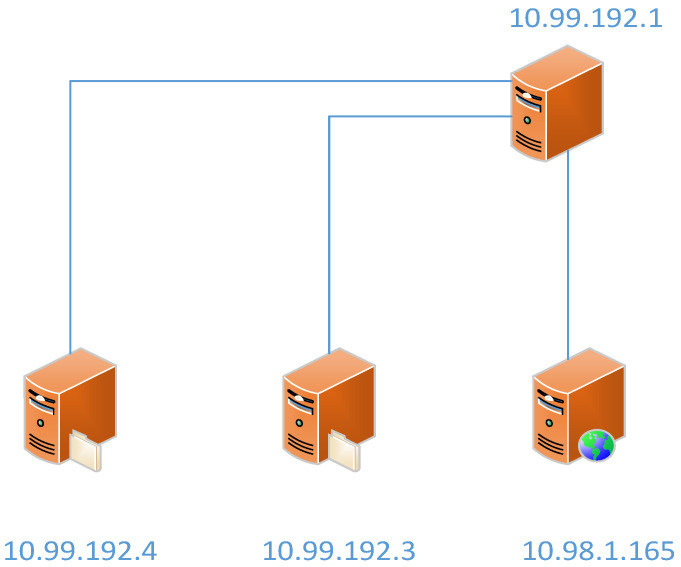
Experimental architecture.

**Figure 4 sensors-23-06139-f004:**
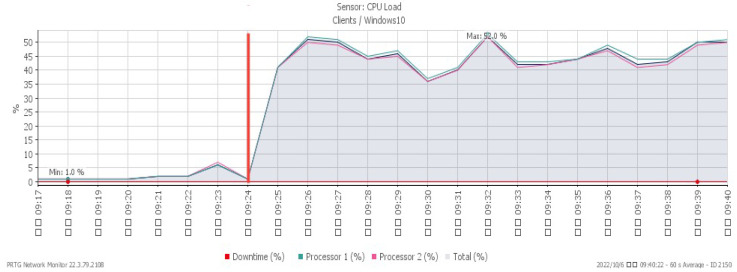
CPU loading under TCP SYN flood.

**Figure 5 sensors-23-06139-f005:**
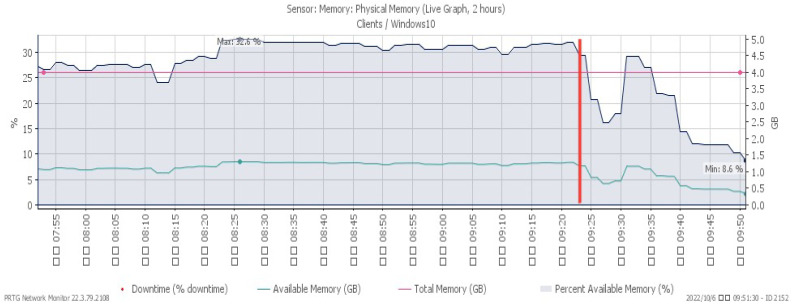
Available memory under TCP SYN flood.

**Figure 6 sensors-23-06139-f006:**
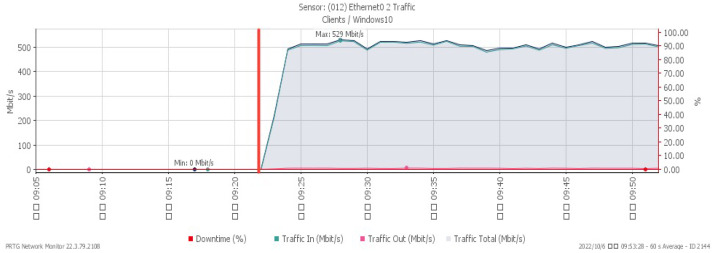
Attack traffic under TCP SYN flood.

**Figure 7 sensors-23-06139-f007:**
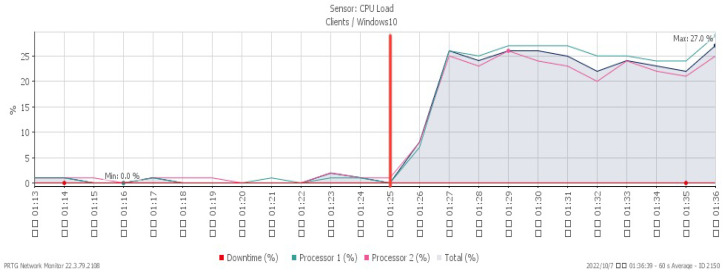
CPU loading under UDP flood.

**Figure 8 sensors-23-06139-f008:**
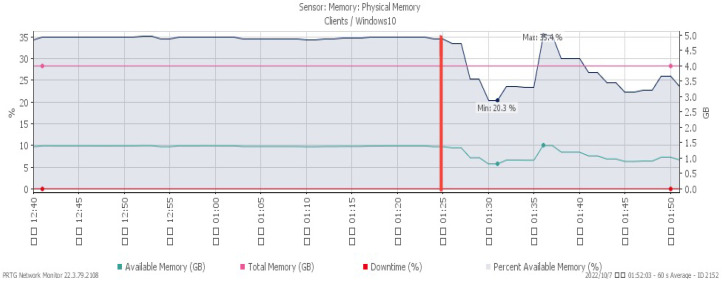
Available memory under UDP flood.

**Figure 9 sensors-23-06139-f009:**
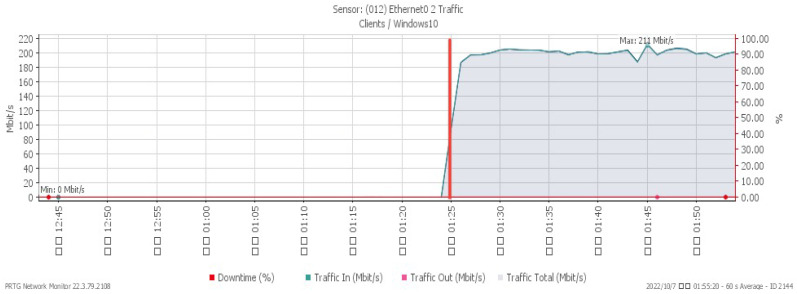
Attack traffic under UDP flood.

**Figure 10 sensors-23-06139-f010:**
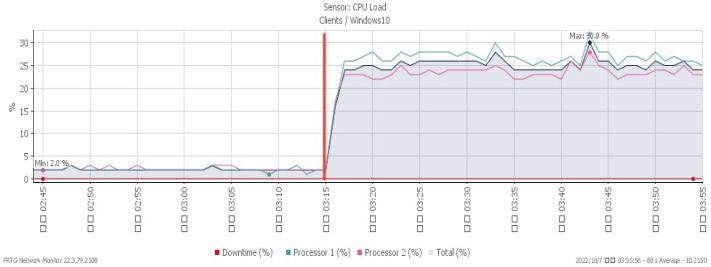
CPU loading under TCP FIN flood.

**Figure 11 sensors-23-06139-f011:**
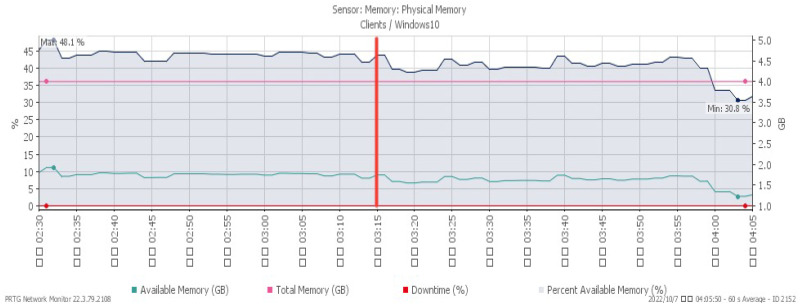
Available memory under TCP FIN flood.

**Figure 12 sensors-23-06139-f012:**
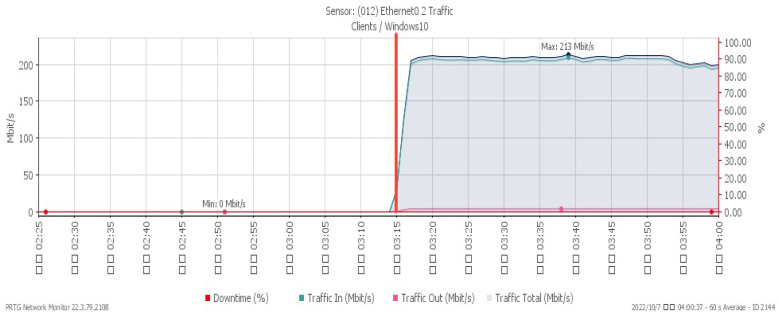
Attack traffic under TCP FIN flood.

**Figure 13 sensors-23-06139-f013:**
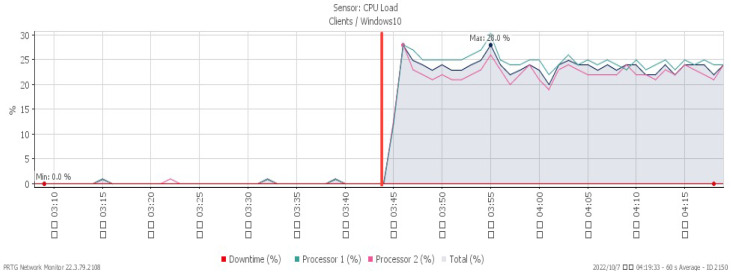
CPU loading under TCP RST flood.

**Figure 14 sensors-23-06139-f014:**
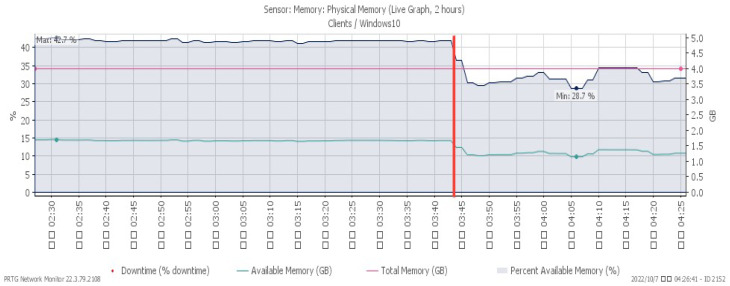
Available memory under TCP RST flood.

**Figure 15 sensors-23-06139-f015:**
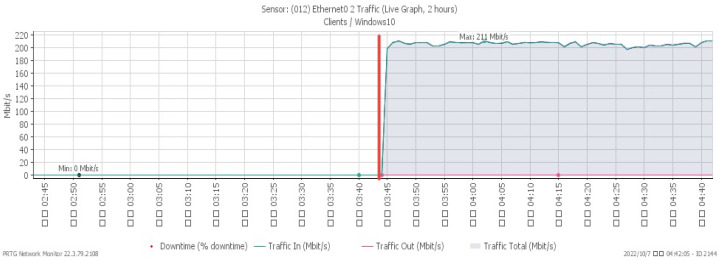
Attack traffic under TCP RST flood.

**Figure 16 sensors-23-06139-f016:**
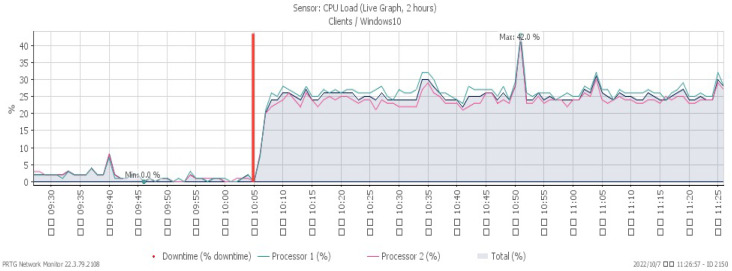
CPU loading under PUSH/ACK flood.

**Figure 17 sensors-23-06139-f017:**
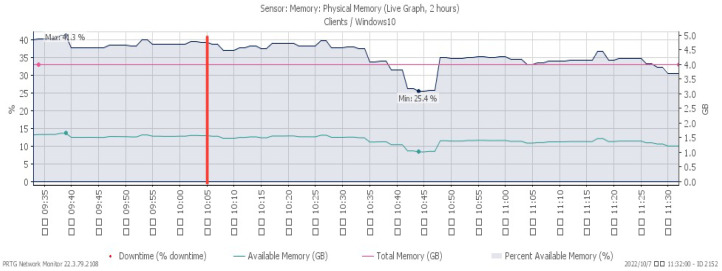
Available memory under PUSH/ACK flood.

**Figure 18 sensors-23-06139-f018:**
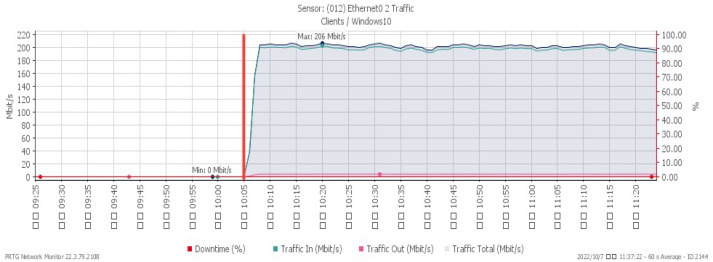
Attack traffic under PUSH/ACK flood.

**Figure 19 sensors-23-06139-f019:**
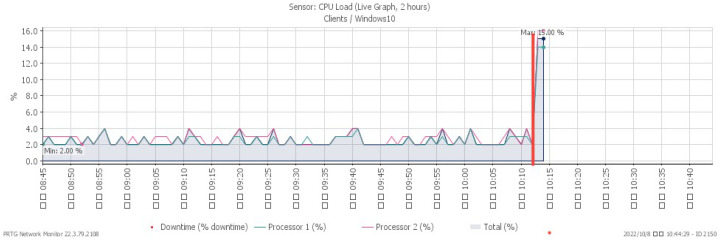
CPU loading under ICMP flood.

**Figure 20 sensors-23-06139-f020:**
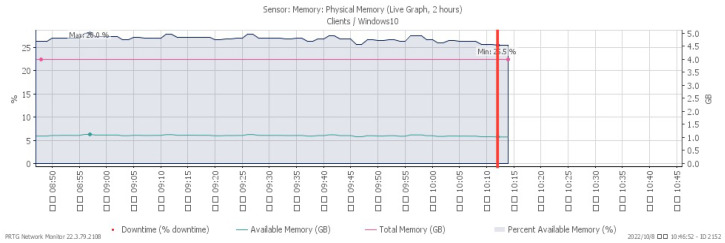
Available memory under ICMP flood.

**Figure 21 sensors-23-06139-f021:**
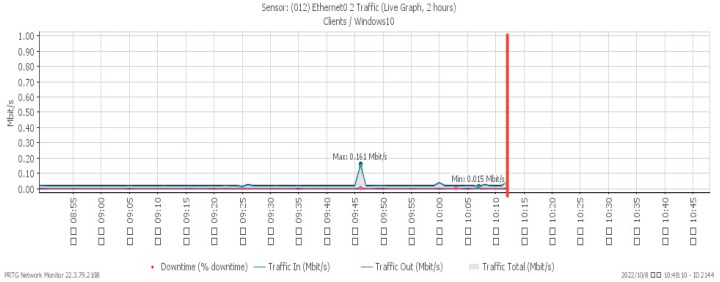
Attack traffic under ICMP flood.

**Figure 22 sensors-23-06139-f022:**
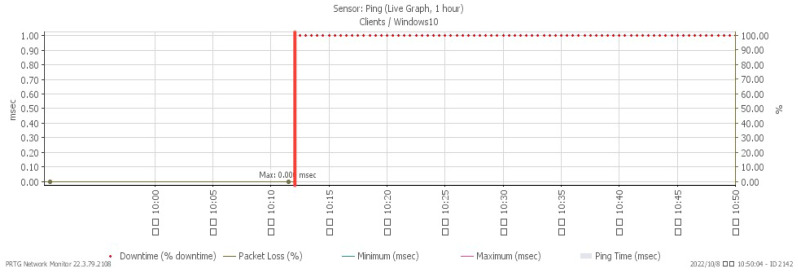
Downtime under ICMP flood.

**Figure 23 sensors-23-06139-f023:**
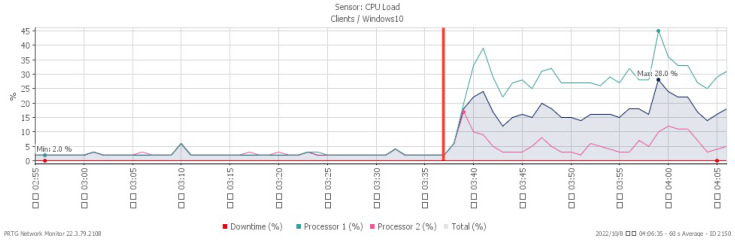
CPU loading under Smurf attack.

**Figure 24 sensors-23-06139-f024:**
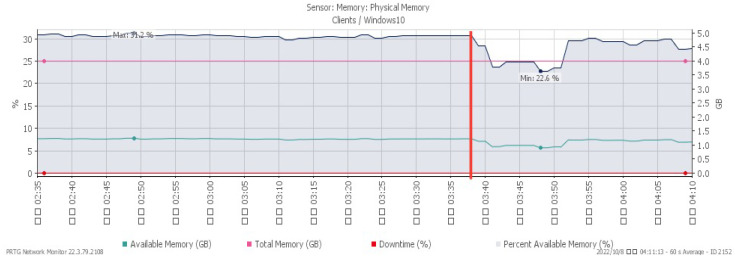
Available memory under Smurf attack.

**Figure 25 sensors-23-06139-f025:**
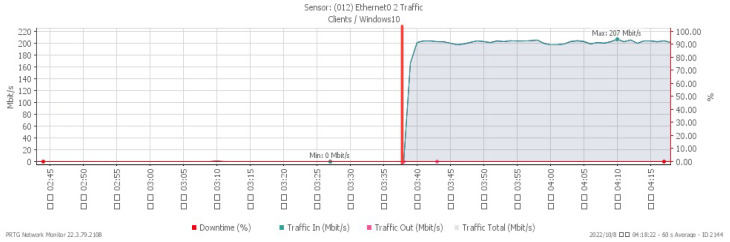
Attack traffic under Smurf attack.

**Figure 26 sensors-23-06139-f026:**
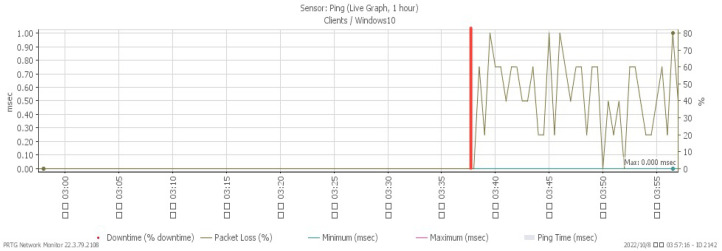
Packet loss under Smurf attack.

**Table 1 sensors-23-06139-t001:** Detailed description of experiment.

Software Spec	Kali 2020.4	Ubuntu 20.04	Ubuntu 20.04	Window10
*Hardware Spec*	2 cores 4 GB Memory	1 core 4 GB Memory	1 core 4 GB Memory	2 cores 4 GB Memory
*Role*	Attack	Defense	Analyzing	Monitoring
*IP address*	10.99.192.1	10.98.1.165	10.99.192.3	10.99.192.4
*Tools*	Hping3	Snort	Wireshark	PRTG

**Table 2 sensors-23-06139-t002:** Comparison of packet loss rate for various scenarios.

	Normal Traffic	TCP SYN Flood	TCP FIN Flood	TCP RST Flood	TCP PUSH and ACK Flood	UDP Flood	ICMP Flood	Smurf
Test 1	0%	11%	1%	1%	0%	0%	100%	100%
Test 2	0%	8%	0%	3%	1%	2%	100%	100%
Test 3	0%	3%	0%	1%	0%	1%	100%	100%
Test 4	0%	3%	0%	1%	0%	1%	100%	100%
Test 5	0%	6%	1%	6%	0%	3%	100%	100%
Test 6	0%	9%	0%	0%	0%	1%	100%	100%
Test 7	0%	8%	0%	0%	0%	1%	100%	100%
Test 8	0%	8%	0%	0%	0%	1%	100%	100%
Test 9	0%	3%	0%	0%	0%	0%	100%	100%
Test 10	0%	5%	1%	0%	0%	0%	100%	100%
Averge	0.0%	6.4%	0.3%	1.2%	0.1%	1.0%	100%	100%

**Table 3 sensors-23-06139-t003:** Comparison of response time for various scenarios.

	Normal Traffic (ms)	TCP SYN Flood (ms)	TCP FIN Flood (ms)	TCP RST Flood (ms)	TCP PUSH and ACK Flood (ms)	UDP Flood (ms)	ICMP Flood (ms)	Smurf (ms)
Test 1	0.24	0.85	0.30	0.30	0.31	0.32	N/A	N/A
Test 2	0.23	0.58	0.38	0.42	0.30	0.27	N/A	N/A
Test 3	0.25	0.93	0.31	0.30	0.31	0.30	N/A	N/A
Test 4	0.24	0.53	0.29	0.33	0.33	0.32	N/A	N/A
Test 5	0.23	0.38	0.29	0.48	0.30	0.35	N/A	N/A
Test 6	0.24	0.49	0.29	0.32	0.39	0.31	N/A	N/A
Test 7	0.24	1.06	0.39	0.31	0.40	0.33	N/A	N/A
Test 8	0.25	0.59	0.30	0.29	0.36	0.36	N/A	N/A
Test 9	0.24	0.41	0.29	0.32	0.33	0.34	N/A	N/A
Test 10	0.23	0.44	0.30	0.31	0.30	0.29	N/A	N/A
Average	0.24	0.63	0.31	0.34	0.33	0.32	N/A	N/A

**Table 4 sensors-23-06139-t004:** Performance comparison between various scenarios.

	CPU Loading (%)	Memory Utilization (%)	Peak Traffic Flow (Mbit/s)	Response Time (ms)	Packet Loss Rate (%)
Normal traffic	1.75	39.40	0.004	0.24	0.0
TCP SYN flood	36.43	39.25	64.73	0.63	6.4
TCP FIN flood	24.71	36.49	69.52	0.31	0.3
TCP RST flood	22.73	36.26	68.33	0.34	1.2
TCP PUSH and ACK flood	24.67	38.76	69.06	0.33	0.1
UDP flood	23.17	36.84	24.83	0.32	1.0
ICMP flood	23.35	54.88	22.27	N/A	100.0
Smurf	20.98	40.48	24.27	N/A	100.0

**Table 5 sensors-23-06139-t005:** Performance comparison before and after attack.

	CPU Loading (%)	Memory Utilization (%)	Peak Traffic Flow (Mbit/s)	Response Time (ms)	Packet Loss Rate (%)
TCP SYN flood	+34.68	−0.15	+64.73	+0.39	+6.4
TCP FIN flood	+22.96	−2.91	+69.52	+0.07	+0.3
TCP RST flood	+20.98	−3.14	+68.32	+0.10	+1.2
TCP PUSH and ACK flood	+22.92	−0.64	+69.06	+0.09	+0.1
UDP flood	+21.42	−2.56	+24.83	+0.08	+1.0
ICMP flood	+21.60	+15.48	+22.26	N/A	+100.0
Smurf	+19.23	+1.08	+24.26	N/A	+100.0

**Table 6 sensors-23-06139-t006:** Performance comparison of the proposed scheme with other methods.

	Accuracy (%)	False Positive Rate (%)	Training Time (s)
The Proposed Scheme	96.13	0.005	N/A
Radial Basis Function Network	94.56	0.01	1320
Support Vector Machine	95.11	0.008	120
Bagging	91.49	0.024	60
J48 Decision Tree	91.82	0.024	7

## Data Availability

Data presented is original and not inappropriately selected, manipulated, enhanced, or fabricated.

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
