# Peer review of "An Experimental Detection of Distributed Denial of Service Attack in CDX 3 Platform Based on Snort"

_sensors, 2023, doi:10.3390/s23136139_

Round 1
Reviewer 1 Report
The core of the manuscript An Experiment of Denial of Service Attacking based on CDX 2 platform is an interesting topic for a broad audience, as it would trigger research from different areas. My feedback on this paper is given below:
- The abstract is required to be more focused to present the contribution of the work. Make the Abstract more concise be explaining the purpose of study; basic study design, methodology and techniques used; major findings; summary of your interpretations, and implications.
-the title of the paper is not adequate; An Experiment of Denial of Service Attacking based on CDX 2 platform--------> An Experimental Detection of Distributed Denial of Service Attack in CDX 2 platform based on SNORT
- In order to represent the citation in the text use one type of notation, either user Nenova et. al. (2019) or [1].
- in line 69, Distributed Denial of Service (DDOS)----> Distributed Denial of Service (DDoS)
- add cloud computing in the list of keywords.
- Give proper references for the statistics presented in lines 26-27
- The flow of paper needs to be improved. Add the Proposed approach section to explain your work by taking a system architecture figure and explain it step by step.
- The result section could be stronger. Add more experimental analysis. Moreover, compare your work with existing methods.
- To have an unbiased view in the paper, there should be some discussions on the limitations of the proposed work
- Intro should be revised , Why we need your study (and it is important), and HOW this contributes to prior literature. I would like to see a clear paragraph directed to theoretical contribution in the introduction. Also, represent the research gap clearly.
- Add the research methodology section to explain the component of proposed approach.
- cite figures in the text and add a proper discussion for them.
- in line 188, author said we use 7 types of DDoS attack generated by Hping3. name the type of DDoS.
- There is no discussion on the cost effectiveness of the proposed method. What is the computational complexity? What is the runtime? Please include such discussions. You can also use the big oh notation to show the computation complexity.
- the author can read the following papers to improve the quality of the research and discussion on the DDoS detection methods:10.1109/TSC.2017.2694426, 10.1109/MENACOMM50742.2021.9678279, 10.4018/IJSWIS.297143
There are some English errors and type errors such as
The second is Intrusion Detection System (IDS), which is Ubuntu-based installed with SNORT------>The second is Intrusion Detection System (IDS), which is Ubuntu-based and installed with SNORT
Reviewer 2 Report
The authors designed an intrusion detection system by monitoring and analyzing the system performance with some tools such as wireshark. In my opinion, some questions must be taken into consideration.
1. There is major difference between the task of preventing the system from attacks and the task of finding a happened intrusion. The former is much more difficult and important than the latter.
2. DDoS is different from DoS. The experiment setup in Figure 1 cannot simulate the distributed attacks.
3. What is the contribution of the work? The authors should develop some algorithms to perform the anomaly detection automatically with better user interface.
4. There are always two squares under the x-axis in the figures 12-34.
Reviewer 3 Report
Summary:
The paper presents an experiment on simulating seven DDoS attack methods and defense experiments on virtual machines on the cloud platform. The attack is detected through packet analysis tools and performance monitoring tools to verify whether the system is under attack. The Intrusion Detection System (IDS) is implemented using SNORT, and alarm records are left when the system is attacked.
Strength:
- The paper presents a practical experiment for detecting and defending against DDoS attacks.
- The use of virtual machines on the cloud platform CDX 3.0 provides an innovative approach to simulate attacks and defense experiments.
- The use of packet analysis tools and performance monitoring tools to identify attacks can provide insight into system performance under different attack scenarios.
Weakness:
- In-depth theoretical analysis or evaluation of the proposed method is not presented.
- How the proposed method performs under different attack scenarios is not discussed.
- No comparison between the proposed method and some widely-known baselines in the field.
- The implementation details for reproducing the method is missing.
Accessible but can be improved.
Reviewer 4 Report
The Abstract in its sub-sections needs re-organization and it does not adequately summarise the gist of the study. The article is proposed to be supplemented with a flowchart illustrating the research technique. A review of the literature is insufficient. It is critical to include some recent work (2018–2020) in the literature review. A literature review should be added in order to illustrate the central topic in a more detailed way. Some further explanations and interpretations are required for the results. It is recommended to include a well-organized discussion of the findings, strengths, and limitations of the present project with additional explanation/details and a conclusion with future work.
The authors should ask for the help of a native English-speaking proofreader because there are some linguistic mistakes that should be fixed.
Round 2
Reviewer 1 Report
The author address all the previous comments but still some issues are pending:
• What motivated you to conduct an experiment specifically on Denial of Service (DDoS) attacks based on the CDX platform? Were there any specific goals or objectives you aimed to achieve through this research?
• Can you provide more details about the configuration and setup of the virtual machines on the CDX platform? How did you ensure that the simulation accurately represented real-world DDoS attack scenarios?
• What criteria or indicators did you use to determine the effectiveness of the defense end (Intrusion Detection System) based on SNORT in detecting and mitigating DDoS attacks? Did you evaluate its performance in terms of accuracy, false positives, false negatives, or any other metrics?
• How did you analyze the traffic flow and packet patterns once the attack was detected? What methods or techniques were employed to distinguish normal network traffic from malicious DDoS attack traffic?
• Author can read the following papers to increase the technical strength of the paper: Optimal Load Distribution for the Detection of VM-Based DDoS Attacks in the Cloud, Attack-Specific Feature Selection for Anomaly Detection in Software-Defined Networks, Distributed denial-of-service (DDoS) attacks and defense mechanisms in various web-enabled computing platforms: issues, challenges, and future research directions.
• Can you elaborate on the role of Wireshark and PRTG at the analyzing end and monitoring end, respectively? How did these tools contribute to the identification and monitoring of DDoS attacks during the experiment?
Minor editing of English language required
Reviewer 2 Report
This time, I have the following comments:
1. The four arguments from line 253 to line 265 are essential for this manuscript. I suggest the author double-check for spelling and other errors before submitting. What does it mean by "Apparently, ……" in line 264?
2. The authors should provide a reasonable analysis of why their method can outperform other methods. Experimentation is important, but it is not enough to make their point.
Reviewer 3 Report
The author addressed all my concerns.
Author Response
Thanks for the reviewer's suggestion.